# Embryo–Uterine Cross-Talk: Exploration of the Immunomodulatory Mechanism in Buffalo

**DOI:** 10.3390/ani12223138

**Published:** 2022-11-14

**Authors:** Lakshmi Devi Huidrom, Shital Nagargoje Dhanaji, Sriti Pandey, Vikash Chandra, Taru Sharma Gutulla

**Affiliations:** 1Division of Physiology and Climatology, ICAR-Indian Veterinary Research Institute, Izatnagar 243122, India; 2Rewa Veterinary College, Nanaji Deshmukh Veterinary Science University, Jabalpur 482001, India; 3National Institute of Animal Biotechnology, Hyderabad 500032, India

**Keywords:** buffalo, embryo, immunomodulation, maternal recognition, embryo development

## Abstract

**Simple Summary:**

Early embryonic loss is the major cause of repeat breeding in buffalo. For successful pregnancy outcomes, proper communication between the embryo and maternal system is very important. When the embryo reaches the uterus, being semi-allogenic in origin, it has to cope from maternal immune reactions to further survive. To survive better in the maternal system, proper embryo functioning is important to give proper signals to the maternal system, and at the same time, an adequate conducive maternal environment is also required for embryonic survival. It was found that blastocyst-stage embryos release some specific molecules (such as interferon-tau), which act as signals to the maternal endometrium. To this response, several genes are modulated in the maternal system and affect expression of several genes in the embryo as well. This way, the embryo survives in the maternal system, and pregnancy continues.

**Abstract:**

Understanding the molecular cross-talk between the embryo and uterine endometrium is crucial for the improvement of IVF outcomes. The present work was undertaken to investigate the effect of pre-implantation embryo on the expression profile of immune-related genes in uterine epithelial cells (UECs) and PBMCs in buffalo. UECs were isolated from slaughterhouse-derived non-gravid uteri, cultured ex vivo and characterized, and buffalo embryos were produced in vitro from slaughterhouse-derived ovaries. Embryos co-cultured with steroid-treated UECs significantly stimulated (*p* < 0.05) the relative mRNA abundance of *PTGS2*, *ISG15*, *OAS1*, *MX2*, *IFNAR1* and *IFNAR2* in UECs while they significantly suppressed the mRNA expression of *NFkβIA*, *NFkβ2*, *TNFα* and *IL1B*, with no significant change in *TGFβ1* and *IL10* in the co-culture of embryos with UECs. In vitro treatment of PBMCs with conditioned media (CM) derived from embryos as well as UEC–embryo co-culture upregulated the mRNA abundance of *ISG15*, *TGFβ1*, *PTGS2OAS1*, *MX2* and *STAT1* while it downregulated *IL17* and *TNFα* expression. The expression of *IFNAR1* and *IFNAR2* was elevated in PBMCs cultured in embryo-derived CM, but there was no significant change in PBMCs cultured in UEC–embryo co-culture CM. Thus, it can be concluded that the developing embryo and its secretions modulate the expression of immune responses by inducing an anti-inflammatory action in uterine epithelial cells for acceptance of the semi-allogenic embryo in the uterus to sustain pregnancy in buffalo.

## 1. Introduction

The uterine epithelium is the area of contact between the embryo and maternal tissue in utero. Both the maternal and embryonic units work together to coordinate dynamic activities in the embryo–maternal relationship. Early embryonic mortality is the major cause of infertility in buffalo [1]. Reproductive efficiency in buffaloes can be improved through better understanding about the key events happening during the early stage of embryonic development. Thus, understanding maternal–embryo cross-talk is necessary to understand and elucidate the causes of pregnancy loss and accordingly design suitable strategies to improve pregnancy outcome in buffalo.

Successful pregnancy establishment and maintenance requires proper maternal recognition of pregnancy (MRP), implantation, placentation and modulation of the maternal immune system that depends on interactions between the conceptus and the maternal system in ruminants. Proper regulation of the complex cellular and molecular relationship between the embryo and the receptive endometrium must be maintained. Important factors regulating pregnancy are progesterone (P_4_) and interferon tau (IFN-τ) [2,3]. Termination of pregnancy is often due to failure of proper communication between the embryo and the maternal system [4]. In cattle, the estimated conception loss occurring mostly during the early stage (from day 8 to 27) of pregnancy is approximately 30%, withpregnancy failure within the first three weeks of pregnancy [5]. Inadequate concentrations of circulating progesterone (P_4_) may be one of the reasons [3]. This may affect downstream expression of endometrial genes and histotroph production into the endometrial lumen [6]. During pregnancy, maternal immunomodulation in ruminants is associated with the action of IFN-τ, the key factor for MRP, and it remains the primary operator of the maternal immune response [2].

During pregnancy, the immune complex may be the important factor regulating pregnancy maintenance, and deregulation of immune responses during conception may be responsible for pregnancy losses [7]. Therefore, synchronous signaling between the embryo and uterine endometrium during the pre-implantation period is crucial for the implantation of the embryo, placentation, and development of the normal embryo [8]. During this period, several immune modifications take place in the endometrium and embryo, which are essential parts for coordinated cross-talk for the establishment of conception [9].

The early-stage embryo is well nourished by histotrophs, a secretion from the intercaruncular endometrium of the uterine glands, and afterward, it forms an adjacent physical interconnection with the endometrial caruncles during implantation (attachment) [10]. Therefore, immunologic pregnancy shows contradiction where the foreign embryo immunologically has the ability to form direct contact with the maternal endometrium that continues throughout pregnancy and protects against pathogens [11]. The embryo modifies the expression of endometrial molecules by suppressing the maternal immune response, thereby supporting the acceptance of the embryo by the maternal immune system. Therefore, the uterus is considered as an immunologically privileged area, as the embryo has the ability to survive in the presence of the maternal immune system [12]. Within the uterus, the semi-allogenic embryo confers tolerance that may be due to the downregulation of nuclear factor kappa B (NFkB) protein (a central mediator of inflammatory and immune responses) in the uterine fluid of bovines on day 8 of pregnancy [13]. Classical polymorphic major histocompatibility complex (MHC) class 1 proteins are not expressed by trophoblast cells in areas in contact with the maternal endometrium in bovines [11]. IFN-τ acts on the luminal epithelium as well as in the superficial glandular epithelium in a paracrine fashion to inhibit oxytocin receptor expression, thereby blocking prostaglandin F2α (PGF2α) luteolytic pulses and thus maintaining the corpus luteum (CL) [14]. On the other hand, IFN-τ inhibits lymphocyte proliferation. Thus, IFN-τ acts as an immunosuppressive factor that may defend the fetus (semi-allogenic) from maternal immune system attack [15]. During the peri-implantation period, the secretion of IFNs triggers the expression of specific genes pertaining to interferon-stimulated genes (ISGs). The major genes found in the peripheral blood leukocytes are interferon-stimulated protein 15 kDa (ISG15), 20–50 oligoadenylate synthetase (OAS-1), and myxovirus-resistance proteins 1 and 2 (MX1 and 2) in dairy cows [16], suggesting that the availability of immune cells in the female reproductive system may also react to IFN-τ. Proper stabilization between anti-inflammatory and pro-inflammatory cytokine molecules is indispensable in maintaining the integrity of the maternal immune system, which will prevent rejection of the embryo for a successful pregnancy [17]. The above literature shows that most of the work related to embryo–uterine cross-talk has been conducted in cattle and sheep, and no work has been conducted in buffalo. Therefore, the present study was aimed at exploring the role of the factors that regulate the modulation of the maternal immune system and the acceptance of the semi-allogenic embryo in the uterus in buffalo, which may ultimately give insights into the interventions for improving animal reproductive performance, especially in buffalo.

## 2. Materials and Methods

All plastic-wares used in this study were cell culture grade and were procured from Nunc (Roskilde, Denmark). Culture media used were cell culture grade, endotoxin tested and procured from Sigma Chemicals Co. (St. Louis, MO, USA). Media and reagents used are described in the texts wherever required. Antibodies were procured from Santa Cruz (SC, Dallas, TX, USA) and Novus Biologicals (NBP, Centennial, CO, USA). The experimental trials were performed in triplicate.

### 2.1. Culture of Uterine Epithelial Cells

#### 2.1.1. Isolation and Culture of Uterine Luminal Epithelial Cells (UECs)

Buffalo uterine epithelial cells were isolated by enzymatic method using 0.25% trypsin-EDTA as per described protocol by Pandey et al. [18]. Briefly, fresh bubaline uteri samples (collected thrice to check repeatability) at early to mid-luteal phase of the estrous cycle were obtained from a local abattoir and washed 3–4 times with normal saline solution (NSS) supplemented with 50 μg/mL gentamycin sulphate solution (G1272, Sigma, St. Louis, MO, USA). Extra tissue was excised, both uterine horns were firmly sealed with cotton thread, and horns were cut from uterus. Uterine lumen of the horns was flushed using phosphate buffer saline (70011-044, Gibco Life Technologies, Waltham, MA, USA) (1× PBS) supplemented with 0.1% bovine serum albumin (BSA) (05482; Sigma) along with 100 μg/mL gentamycin twice to eliminate all the debris from the uterine lumen. After proper flushing, the uterine lumen was infused with 10–15 mL of 0.25 percent trypsin-EDTA (T4799, Sigma; E6758, Sigma), the horn was tied with cotton thread from the other side and incubated at 38.5 °C for 60 min in shaker incubator (Matrix Eco Solution, Delhi, India) with 70–100 moves/min. After incubation, each horn was tilted in order to detach the cells. Uterine contents in suspension were aspirated out from the uterine lumen with 10 mL syringe and 18-gauge needle. The collected cell suspension was centrifuged for 10 min at 250× *g*. Cell pellets were resuspended in high-glucose Dulbecco’s modified eagle medium (DMEM) (D5796, Sigma) containing 10% FBS (Lot# 2106467RP, Gibco) and antibiotic–antimycotic solution (Lot# 2257208, Gibco) at 20 μL/mL concentration. Then, it was centrifuged at 250× *g* for 6 min (twice) to ensure the pellet was washed properly. The supernatant was discarded, and 1 mL of RBC lysis buffer (R7767, Sigma) was added to the cell pellet and kept for 2–3 min at 37 °C, if RBCs are present. Again, the solution was centrifuged at 250× *g* for 5 min to give final washing, and the pellet was resuspended in culture media containing 10% DMEM. For final washing, the solution was centrifuged again for 5 min at 250× *g*. Thereafter, the pellet was resuspended in 10% DMEM and the solution strained with cell strainer of 70 μm mesh size (352350, BD Falcon ™, Arizona, United States of America). Cell concentration and viability were evaluated with 0.4% trypan blue dye (T8154, Sigma) with the aid of a Neubauer’s hemocytometer chamber under phase contrast microscope. Finally, cells were seeded in 24-well plate (142475, Thermo Scientific, Waltham, MA, USA) at a concentration of 5 × 10^4^ cells/cm^2^ and incubated in 5% CO_2_ incubator at 38.5 °C with maximum relative humidity. After 24 h of culture, along with the media, the uterine luminal epithelial cells (UECs) were collected and re-plated into a new 24-well plate. After every 48 h, the culture media were changed until reaching 70–80% cell confluence. After reaching 70–80% confluence, cells were passaged using Accutase^®^ (A6964; Sigma) and reseeded at 5 × 10^4^ cells/cm^2^.

#### 2.1.2. Characterization of Uterine Luminal Epithelial Cells (UECs)

The characterization of uterine luminal epithelial cells (UECs) was determined according to the cell attachment rate, its phenotypic characteristics and expression of specific cell markers. Cell attachment was observed under inverted microscope (Olympus, Tokyo, Japan) at every 6 h interval. Phenotypic characterization was evaluated by cell shape, compactness as well as growth pattern of monolayer in primary culture. Localization of uterine luminal epithelial cell positive marker ‘cytokeratin’ along with negative marker ‘vimentin’ was also performed by immunocytochemistry (ICC) and PCR assay [19].

#### 2.1.3. Immunocytochemistry (ICC)

Immunocytochemistry of uterine epithelial cells was performed following protocol of Pandey et al. [18]. Briefly, primary culture of UECs monolayer attaining confluency of 70–80% was fixed with 4% formaldehyde for 20 min at 37 °C. Thereafter, it was permeabilized with 0.25% Triton-X for 15 min at 37 °C, and nonspecific binding was blocked by treating with 5% BSA (8806; Sigma) for 45 min. The cells were probed with anti-cytokeratin 5/8 antibody (Santa Cruz Biotechnology, Cat#SC-32328) and anti-vimentin antibody (Novus Biologicals, Cat#NBP1-31327) at 1:100 dilutions and incubated overnight at 4 °C. They were further treated with goat anti-rabbit antibody conjugated with phycoerythrin (PE) (sc-3739) and donkey anti-goat conjugated with Texas red (TR) (sc-2785) at dilution rate of 1:200 for 2 h at 37 °C in a dark environment and thereafter counterstained with 4′,6-diamidino-2-phenylindole (DAPI) (Santa Cruz; SC-3598) to stain the nuclei of the cells. Similarly, negative controls were processed, except omission for primary antibodies and images was taken under fluorescent microscope (IX 71, Olympus, Shinjuku, Tokyo, Japan).

#### 2.1.4. Treatment of Epithelial Cells with Steroid Hormones

Characterized uterine epithelial cells (UECs) monolayer at first passage (P1) was exposed to steroid hormones (estradiol-17β and progesterone) to mimic the internal milieu of estrous cycle. Initially, it was treated with estradiol-17β at a concentration of 10 pg/mL for 24 h following culture media supplemented with estradiol-17β at a rate of 5 pg/mL and progesterone at a rate of 3.14 ng/mL for 5 consecutive days on alternate days [3,18].

#### 2.1.5. In Vitro Embryo Production

In vitro embryo production was performed as per earlier described protocol given by Bhardwaj et al. and Pandey et al. [20,21]. Initially, buffalo ovaries were procured from local abattoir in pre-warmed normal saline solution (NSS) and washed thoroughly with normal saline solution. All the visible follicles were aspirated using 18-gauge needle. The aspirated cumulus oocyte complexes (COCs) were categorized according to their morphological appearance and uniform granular ooplasm [20]. The COCs of grade A and B were subjected to in vitro maturation (IVM) in maturation media comprising TCM 199 (HEPES modified) supplemented with 5 μg/mL LH, 0.5 μg/mL FSH, 1 μg/mL estradiol-17β, 0.25 mM sodium pyruvate, 0.68 mM L-glutamine, 10% FBS, 10 μg/mL gentamicin and 3 mg/mL BSA and incubated in 5% CO_2_ at 38.5 °C for 24 h with maximum relative humidity. In vitro fertilization (IVF) was processed using Tyrode–albumin lactate–pyruvate (FerTALP) media supplemented with 20 μg/mL heparin, 0.2 mM sodium pyruvate, 6 mg/mL BSA (fatty acid free) along with frozen thawed buffalo bull semen. The matured COCs (10–15) were kept for co-incubation in processed semen droplets of 50 μL in 5% CO_2_ at 38.5 °C for 18 h with maximum relative humidity. After 18 h co-incubation, presumptive zygotes were transferred to modified synthetic oviductal fluid (mSOF) supplemented with 0.25 mM sodium pyruvate, 0.68 mM L-glutamine, 3 mg/mL BSA (fatty acid free), 50 μg/mL gentamycin sulphate along with 1% non-essential and essential amino acids. Subsequently, embryos were cultured in modified synthetic oviductal fluid (mSOF) supplemented with 10% fetal bovine serum (FBS).

### 2.2. Experiment 1: Effect of Pre-Implantation Embryo on the Expression Profile of Immune-Related Genes in Uterine Epithelial Cells

#### 2.2.1. Co-Culture of Embryos with Steroid-Treated UECs

Morula stage embryos or day 4 cleavage embryos (*n* = 15 in each group) were co-cultured with steroid-treated UECs (as per “Section 2.2”, i.e., initially, it was exposed to estradiol-17β at a concentration of 10 pg/mL for 24 h following that, the culture media were changed with media supplemented with estradiol-17β at a rate of 5 pg/mL and progesterone at a rate of 3.14 ng/mL for 5 days on alternate days [18] on luminal epithelial cells monolayer (treatment, T) and incubated for 4 days. Steroid-treated luminal epithelial cells without embryo were taken as control (Table 1). UECs were collected from both groups without embryos, and total RNA was isolated for gene expression studies. The experimental groups are given below.

#### 2.2.2. Real-Time Polymerase Chain Reaction (RT-PCR)

Total RNA was isolated from uterine epithelial cells (UECs) and blastocysts using Trizol reagent. The total RNA purity and concentration were checked by Nanodrop Spectrophotometer and read at OD260 and OD280 taken against 1 μL nuclease free water as blank. Further, for cDNA synthesis, samples with OD260:OD280 values between 1.8 and 2.0 were used. Reverse transcription (RT) was performed in a total 20 μL reaction volume using the Verso cDNA synthesis kit (AB-1453/B; Thermo Scientific, USA) following the manufacturer’s instructions. A sum of 1 μg of total RNA was used in the RT as a template. The PCR amplification was carried out for selected genes using Platinum PCR supermix (11306-016; Life Technologies, USA) as per the manufacturer’s instructions. Details of the genes, its specific annealing temperature, the sequence of sense and anti-sense primer and product length are listed in Table 2. Amplification for specific RT-PCR product was confirmed by 1.8% agarose gel electrophoresis. Quantitative real-time PCR was performed using DyNAmo SYBR green (F-416L; Thermo Scientific, USA) and BioRad CFX real-time system.

#### 2.2.3. Gene Expression Analysis

Steroid-treated luminal epithelial cells without embryos were used as a calibrator for obtaining the expression profiles of immune-related genes by relative mRNA expression. Glyceraldehyde-3-phosphate dehydrogenase (GAPDH) was used as the housekeeping gene. Efficiency-corrected relative quantification of mRNA was obtained as described earlier [22]. For this, efficiency of primers was determined by serial dilution of template cDNA sample and was run in triplicate.
(1)$$Ratio=\frac{E_{target}{}^{\Delta Ct\;target\;\left(\text{control-sample}\right)}}{E_{ref}{}^{\Delta Ct\;ref\;(\text{control-sample})}}$$
where Ratio is the relative expression, E_target_ is the real-time efficiency of the target gene transcript, and E_ref_ is the real-time efficiency of the housekeeping gene transcript.

### 2.3. Experiment 2: The Effect of Conditioned Media (CM) from Embryos and UEC–Embryo Co-Culture in PBMCs

#### 2.3.1. Collection of CM from Embryos without UECs

As per the above protocol given in Section 2.1.5, the collection of oocytes, IVM, IVF and in vitro embryo culture was performed accordingly. Briefly, presumptive zygotes, 18 h post IVF, were cultured in mSOF without serum. After 48 h, media were again changed while culturing 20 zygotes per 400 µL of mSOF supplemented with 10% FBS in the 4-well plates with mineral oil, incubated at 38.5 °C and 5% CO_2_ with maximum humidity, and subsequently, media were changed after each 48 h [23]. On the 5th day of embryo culture, media were changed with 5% FBS, left for a further 72 h, and conditioned media were collected and stored at −20 °C.

#### 2.3.2. Collection of CM from Steroid-Treated UEC Co-Cultured Embryos

Steroid-treated UEC cultured media were completely replaced with mSOF supplemented with 10% FBS before adding embryos. Then, 66 h post IVF, 20 zygotes were transferred on steroid-treated UECs monolayer and incubated at 38.5 °C and 5% CO_2_ with maximum humidity. Media were changed every alternate day. On the 5th day, media were changed with 5% FBS and kept for 72 h. Conditioned media were collected and stored at −20 °C.

#### 2.3.3. Isolation of PBMCs

Blood samples (10 mL) were collected in heparinized vacutainers after puncturing the jugular vein from healthy nonpregnant buffalo. PBMCs were isolated by density gradient centrifugation method by using Histopaque (10771, Sigma) as per manufacturer’s instruction; briefly, the whole blood was diluted with 1× PBS and gently overlayered on Histopaque. The opaque layer between the plasma and Histopaque was pipetted off into another 15 mL centrifuge tube. The isolated PBMCs were washed thrice with 1× PBS and centrifuged at 425× *g* for 5 min. All the steps for PBMC isolation were performed at room temperature as per manufacturer’s instructions. Cell viability and concentration were evaluated using 0.4% trypan blue with the aid of Neubauer’s hemocytometer.

#### 2.3.4. Treatment of PBMCs with CM

The above isolated PBMCs were cultured with conditioned media obtained from embryos alone (treatment 1) and conditioned medium obtained from steroid-treated UECs co-cultured with embryos (treatment 2). PBMCs co-cultured with plain embryo culture media were taken as control (C). PBMCs were harvested from respective treatment groups at 48 h, and gene expression study was conducted.

#### 2.3.5. Total RNA Isolation and RT-PCR Assay

The total RNA was isolated from PBMCs by Trizol method, purity of RNA was checked, and cDNA was prepared. Primers were synthesized using Genetool software or obtained from available literature and custom synthesized, and qPCR analysis of immune-related genes was performed as per the methodology described earlier.

#### 2.3.6. Gene Expression Analysis

The relative mRNA expression was studied as per the protocol described in Section 2.2.3.

### 2.4. Statistical Analysis

All experimental data are shown as mean ± SEM. The statistical significance of mRNA expression for immune-related genes studied was assessed using SAS 9.2 software (SAS Institute Inc., Cary, NC, USA) by one-way ANOVA with Tukey’s post hoc test using graph-pad Prism V 5.0 software. Differences were considered significant if *p* < 0.05.

## 3. Results

Agarose gel electrophoresis demonstrated a clear band of 184 bp of IFNτ (figure not shown), indicating the transcription of IFNτ in the developing blastocyst.

### 3.1. Isolation and Characterization of Uterine Epithelial Cells

UECs were successfully isolated using 0.25% trypsin–EDTA. UECs were characterized based on the cell attachment rate, phenotypic characterization and immunolocalization along with mRNA expression of specific cell markers of UECs. UECs started attachment over the culture plate after 24 to 48 h as a single or clumps of variable sizes. Thereafter, epithelial cell clumps started to expand, rapidly forming a colony on the culture plate. It took about 7–8 days to attain 70–80% confluence, which showed typical cuboidal cell morphology with a tight and compact monolayer in the primary culture (Figure 1).

Purity of UECs was analyzed by immunocytochemistry (ICC) by using specific markers. The fluorescence staining with cytokeratin, which is a positive marker, was evident in epithelial cells (Figure 2). Anti-vimentin antibody fluorescence was absent, indicating the absence of UEC negative marker (Figure 2). The purity of the cultured uterine epithelial cells was >98%.

The mRNA expression depicted the presence of a bright band on the agarose gel, of which electrophoresis evidenced the existence of a UEC positive marker, cytokeratin (147 bp), and no band in the vimentin lane (163 bp).

### 3.2. Effect of Embryos on Expression of Immune-Related Genes in UECs When Co-Cultured with Steroid-Treated UECs

When embryos were co-cultured with steroid-treated UECs, they significantly (*p* < 0.05) stimulated the relative mRNA abundance of *ISG15*, *OAS1* and *MX2* (Figure 3). Similarly, *STAT1* along with *IFNAR1* and *IFNAR2* were significantly (*p* < 0.05) upregulated in UECs co-cultured with embryos (Figure 3). However, the co-culture of embryos in UECs significantly (*p* < 0.05) downregulated the mRNA expression of *NFkβIA*, *NFkβ2*, *TNFα* and *IL1B* (Figure 4), and no significant (*p* > 0.05) change in the expression of *TGFβ1* and *IL10* was observed (Figure 4). The expression of *PTGS2* was significantly (*p* < 0.05) increased in UECs (Figure 4).

### 3.3. Effect of Conditioned Medium (CM) from Embryos and UEC Co-Cultured Embryos on Expression of Immune-Related Genes in PBMCs

The mRNA abundance of transcripts *ISG15*, *OAS1*, *MX2* and *STAT1* in PBMCs of treated groups (embryos alone, T1 and steroid-treated UECs co-cultured embryos, T2) were significantly (*p* < 0.05) upregulated as compared to control in PBMCs (Figure 5). However, a non-significant (*p* > 0.05) change was observed between T1 and T2 groups. Additionally, *IFNAR1* and *IFNAR2* expression was significantly (*p* < 0.05) elevated in T1 compared to T2 and control group (Figure 5). The expression profile of *NFKIA*, *NFKβ2*, *IL10* and *IL1B* mRNA showed no significant (*p* > 0.05) change between T1, T2 and control groups (Figure 6). *TGFβ1* and *PTGS2* mRNA expression was significantly upregulated (*p* < 0.05) in T1 and T2 as compared to control, but insignificant (*p* > 0.05) changes were observed between T1 and T2 (Figure 6), while significant downregulation of both the transcripts, i.e., *IL17* and *TNFα* (*p* < 0.05), was observed in both T1 and T2 as compared to control (Figure 6).

## 4. Discussion

Successful pregnancy needs the acceptance of a semi-allogenic embryo/fetus by the maternal immune system, which is accomplished by a sequence of complex interactions between the developing embryo and the maternal system. The present study provides strong evidence for the existence of immunological cross-talk between UECs and early pre-implantation embryos in buffalo. Furthermore, the in vitro embryos can modulate the expression of different immune-related genes for the acceptance of semi-allogenic embryos during pregnancy in buffalo. Although, protein-expression-based studies are best for advocating the action of biomolecules, but due to some limitations, this study is performed entirely on the basis of expression-based mRNA.

In our experiment, we used 20 embryos to co-culture them with the steroid-treated UECs monolayer, for the simulation of a stimulus from embryo-derived *IFNτ*. This was clearly evident from the modulation of gene expression in UECs and PBMCs. Multiple embryo transfers have already been used for studying very early maternal–embryo interactions in vivo in cattle [24,25].

Our results showed that the co-culture of embryos with steroid-treated UECs significantly upregulated the relative mRNA expression of *ISG15*, *OAS1*, *MX2*, *STAT1* and type-1 IFN receptors (*IFNAR1* and *IFNAR2*) as compared to control, which was in agreement with earlier findings [15,23]. Embryos from the morula (day 5) to the blastocyst (day 9) stage produce higher amounts of *IFNτ* in bovine uterine epithelial cells, and they eventually can activate an interferon-signaling cascading mechanism [14,15]. In cattle, it is also reported that as early as day 13 of pregnancy, the embryo stimulates *ISGs* expression in the endometrium [26]. Embryo-induced changes in gene expression in the bovine endometrium in the uterus and oviduct both in vivo and in vitro suggest that in cattle, MRP could occur around day 8 of pregnancy [15,23], which may be the case in buffalo.

*IFNτ* is the key factor responsible for MRP, which involves suppression of the endometrial luteolytic process to sustain the corpus luteum for progesterone production through stimulation of the oxytocin receptors of the endometrium epithelia of the uterus by prohibiting the production of *PGF2α* (prostaglandin F2 alpha) pulses (luteolytic) in bovines [2,27]. In the *IFNτ* released from blastocyst, stimulated several genes (ISGs) in the endometrium in cell-specific manor, which are possibly essential for elongation of conceptus, implantation, and pregnancy establishment [28]. In earlier studies, it has been reported that embryos significantly down regulated *NFkβ2*, *NFkβIA* expression, which is a transcriptional factor for inflammatory and immune responses with upregulation of *PTGES* expression (an enzyme related to *PGE2* synthesis) in a bovine uterus [15] and oviduct [15,25]. Additionally, it was reported that in the uterus, the embryos significantly suppressed *IL1B* and *TNFα* (pro-inflammatory cytokines) in bovines [15]. Our data also corroborate with the previous findings that the expression of *NFkβ2*, *NFkβIA*, *IL1B* and *TNFα* transcripts was significantly suppressed when embryos were co-cultured with steroid-treated epithelial cells. The embryos suppressed *NFkβ2* and *NFkβIA* expression (major cytokine for the initiation of inflammatory and immune responses) along with down regulation of *IL1B* and *TNFα* expression (pro-inflammatory cytokines). *IFNτ* acts as an anti-inflammatory agent by suppressing the *NFkβ* mechanism and inhibits production of *IL1B* and *TNFα* in the endometritis in mice [29]. Thus, a decrease in pro-inflammatory cytokines expression is likely due to downregulated *NFkβ* pathways in uterine epithelial cells (UECs) through embryo-derived *IFNτ* [25,30].

It has been reported that prostaglandin E synthase (*PTGES*) expression was stimulated in bovine UECs by embryos [15], which is in agreement with our findings. In cattle, on day 6–7 of pregnancy, prostaglandin endoperoxide synthase (*PTGS*) expression in the uterus was reported to be enhanced in existing viable embryos [31,32]. *PTGS2* is present in the stroma, uterine epithelium and glands of the endometrium at the implantation sites, which is triggered by the developing embryo in the uterine endometrium [33]. *PTGS2* is mostly triggered by growth factors, cytokines and hormones [34]. It initiates the process of biosynthesis through a different family of bioactive lipid mediators known as PGs, such as *PGD2*, *PGE2*, *PGF2α*, *PGI2* and thromboxane (TxA2), in a cell-type restricted fashion [35]. *PTGS2* is also known to be essential for implantation, corpus luteum development, fetus growth and development [36]. Thus, the expression of *PTGES* along with the secretion of *PGE2* was stimulated in UECs by early-developing embryos [15]. Throughout pregnancy, *PGE2* is secreted at the maternal–fetal junction from both the maternal endometrium and the conceptus, which may be a significant factor as an immunomodulatory agent to shield the semi-allogenic fetus from maternal immunological attack in cows and ewes [37,38].

Therefore, the present study suggests that embryo-derived *IFNτ* might contribute to the stimulation of *PTGES* secretion from UECs. Hence, our results strongly advocate that *IFNτ* is the main factor that is produced by the developing embryo in the uterus. The *IFNτ* signal is received by UECs, which induces an alteration of immune responses generating an anti-inflammatory action to the embryo.

Conditioned media (CM) from embryos (T1) and embryos co-culture with steroid-treated UECs (T2) in PBMCs stimulated *ISG15*, *MX2*, *OAS1* and *STAT1* mRNA expression as compared to control. Our findings are similar with earlier studies reported on bovines [15]. Our data could suggest that the conditioned media collected from the embryos and embryos co-cultured with UECs contained conceptus-derived *IFNτ*, which affects downstream signaling pathways to induce *ISGs*. In cattle, it was reported that the transcripts of *ISG15* and *OAS1* expression in PBMCs were higher in pregnant animals than in non-pregnant animals on day 8 of artificial insemination (AI) [39]. These findings could also suggest that in cows, the response of immune cells to the conceptus might be essential to establish pregnancy [39]. In addition, uterine flushing (UF) from pregnant cows on day 7 onward induced *ISG15* and *OAS1* expression in PBMCs, which indicated that embryo-derived *IFNτ* present in uterine flushing of pregnant cows can stimulate interferon-stimulated genes [40]. Therefore, our results strongly advocate that buffalo embryo meditates its immune cross-talk with immune cells and UECs to maintain pregnancy.

Furthermore, the relative mRNA expression of *IFNAR1* and *IFNAR2* in PBMCs was higher in CM from embryos (T1) than CM from the co-culture of embryo–UECs (T2) and control, which was corroborated with earlier findings on bovines [15]. These results suggest that higher IFNτ was present in the conditioned media (CM) from embryo culture (T1) than that of CM from the co-culture of embryo–UECs (T2). Since *IFNτ* binds to type-1 interferon receptors in UECs, it is possible that a small amount of *IFNτ* is present in CM from embryos than from the co-culture of embryo–UECs in bovines [15]. Likewise, the relative abundance of pro-inflammatory cytokines such as *TNFα* and *IL17* was downregulated, whereas the transcripts of *TGFβ1* and *PTGS2* were elevated in CM from embryo (T1) and embryo co-cultured with UECs (T2) in PBMCs as compared to control. However, there were no significant changes noted in *IL10*, *IL1B*, *NFKβ1A* and *NFKβ2* expression in the T1, T2 and control groups. Our data corroborate with earlier findings on bovines [15]. In cows, the transcripts of *TNFα* and *IL1B* (pro-inflammatory cytokines) expression were decreased, while the relative abundance of *IL10* (an anti-inflammatory cytokines) was enhanced in PBMCs during pregnancy [39,40].

Embryos stimulate the expression of *TGFβ1* and *PTGES* in PBMCs to induce an anti-inflammatory role and immune-tolerance conditions in the uterus. In PBMCs, the embryo stimulates *TGFβ1* and *PTGE2* expression, which may function synergistically to induce differentiation of naïve T cells (Th0) to regulatory T cells for immune-tolerance conditions in the uterus [41]. Thus, conceptus-derived factors play an important role in the modulation of the maternal immune system during pregnancy to accept the semi-allogenic fetus.

## 5. Conclusions

The findings from the present study provide strong evidence for the existence of immunological cross-talk between UECs and early pre-implantation embryo in buffalo. The developing embryo produces *IFNτ* and modulates maternal immune response by the upregulation of anti-inflammatory molecules and downregulation of proinflammatory molecules in uterine epithelial cells for the acceptance of the semi-allogenic embryo in the uterus to sustain pregnancy.

## Figures and Tables

**Figure 1 animals-12-03138-f001:**
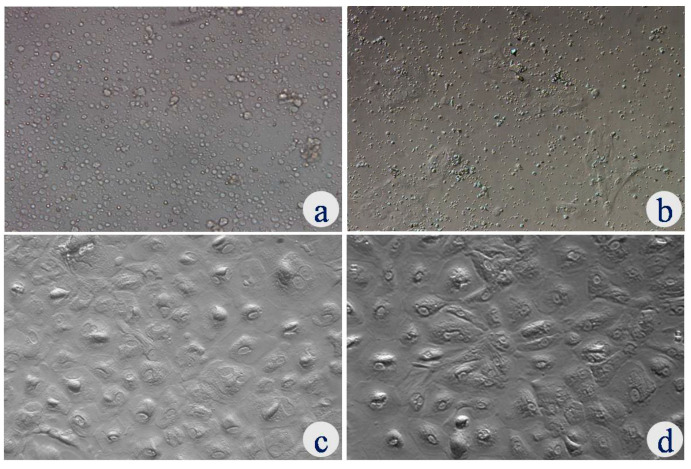
Uterine epithelial cell morphology under phase contrast microscopy: (**a**) isolated epithelial cells seeded on culture plate on day 0; (**b**) epithelial cell clusters on day 2; (**c**) primary epithelial cell culture on day 7–8 (cuboidal to columnar); (**d**) first passage epithelial cells. Magnification—20×.

**Figure 2 animals-12-03138-f002:**
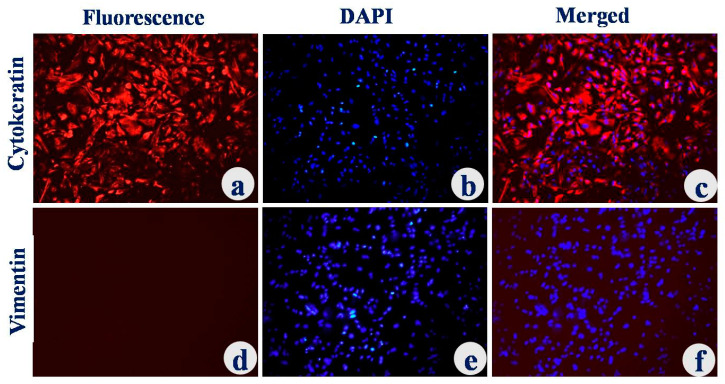
Immunelocalization of cytokeratin and vimentin for characterization of the epithelial cells: Cells were stained with primary antibodies directed against cytokeratin (**a**), vimentin (**d**) and counterstained with secondary antibodies conjugated with Texas red. DAPI-stained nuclei for respective groups (**b**,**e**). Merged photographs of DAPI and secondary antibody (**c**,**f**). Magnification—20×.

**Figure 3 animals-12-03138-f003:**
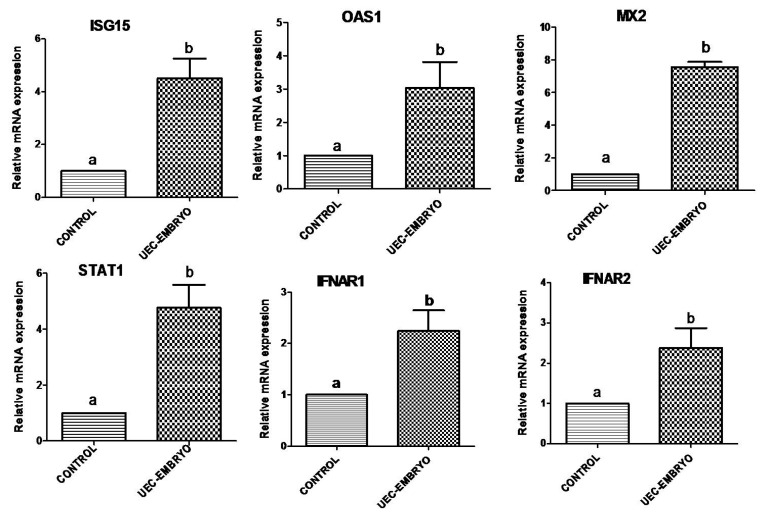
Relative mRNA expression profile of immune-related genes (*ISG15*, *OAS1*, *MX2*, *STAT1*, *IFNAR1* and *IFNAR2*) in steroid-treated uterine epithelial cells co-cultured with embryos (treatment) and without embryos (control). Bars bearing different superscripts differ significantly (*p* < 0.05).

**Figure 4 animals-12-03138-f004:**
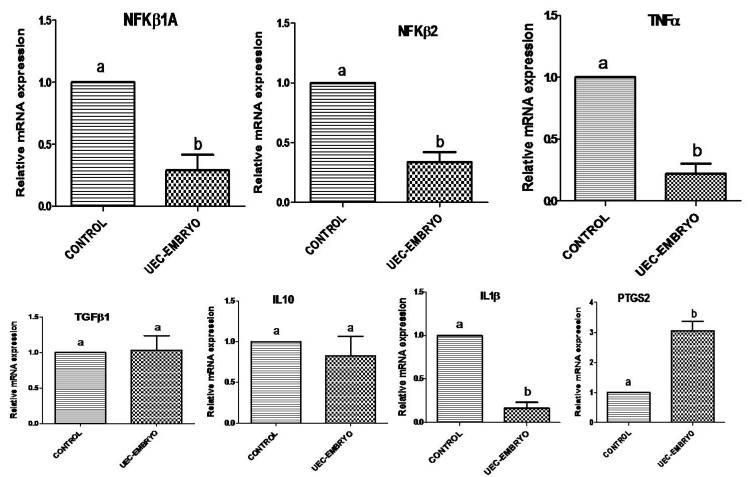
Relative mRNA expression profile of immune-related genes (*NFKβ1A*, *NFKβ2*,*TNFα*, *TGFβ1*, *IL10*, *IL1β* and *PTGS2*) in steroid-treated uterine epithelial cells co-cultured with embryos (treatment) and without embryos (control). Bars bearing different superscripts differ significantly (*p* < 0.05).

**Figure 5 animals-12-03138-f005:**
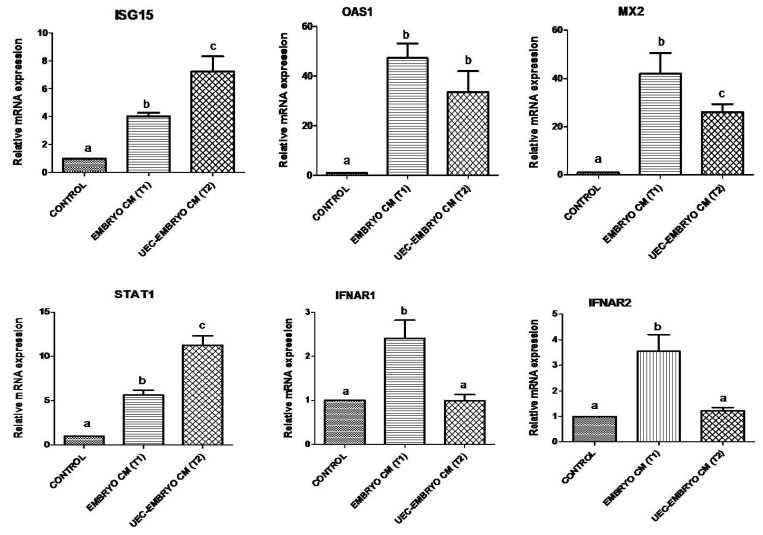
Relative mRNA expression profile of immune-related genes (*ISG15*, *OAS1*, *MX2*, *STAT1*, *IFNAR1* and *INFAR2*) in PBMCs cultured over CM derived from steroid-treated UECs co-cultured with embryos and embryo-cultured media with respect to control. Bars bearing different superscripts differ significantly (*p* < 0.05).

**Figure 6 animals-12-03138-f006:**
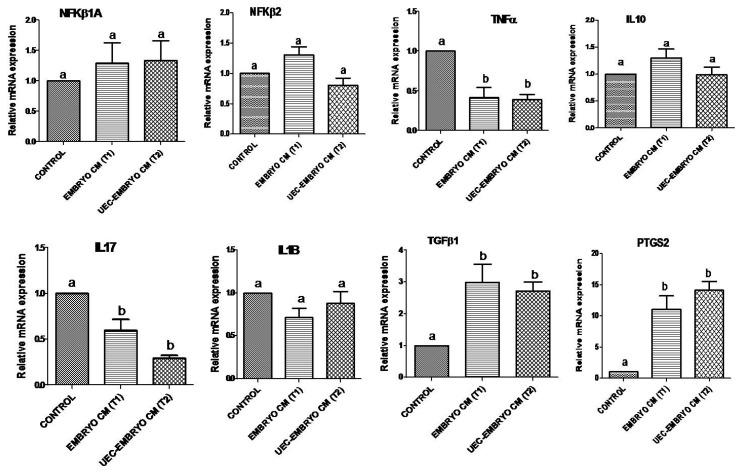
Relative mRNA expression profile of immune-related genes (*NFKβ1A*, *NFKβ2*,*TNFα*, *IL10*, *IL17*, *IL1B*, *TGFβ1* and *PTGS2*) in PBMCs cultured over CM derived from steroid-treated UECs co-cultured with embryos and embryo-cultured media with respect to control. Bars bearing different superscripts differ significantly (*p* < 0.05).

**Table 1 animals-12-03138-t001:** Experimental groups.

Group	Details
Control	Steroid-treated luminal epithelial cells without embryo
Treatment (T)	Day 4 cleavage embryos + Steroid-treated luminal epithelial cells monolayer

**Table 2 animals-12-03138-t002:** Gene-specific oligonucleotide primers for PCR amplification.

Gene	Primer Sequence (5′-3′)	Annealing Temperature (°C)	Amplicon Size (bp)	Accession No.
IFNτ	F-GATGGCCCTGGTGCTGGTCA	58	184	[3]
	R-GTCGCCCTCCACCATCTCCTG			
ISG15	F-TCTGAGGGACTCCATGACGG	55	51	NM_174366
	R-TTCTGGGCGATGAACTGCTT			
OAS1	F-TAGGCCTGGAACATCAGGTC	57	105	NM_178108
	R-TTTGGTCTGGCTGGATTACC			
MX2	F-CTTCAGAGACGCCTCAGTCG	56	232	NM_173941
	R-TGAAGCAGCCAGGAATAGT			
STAT1	F-CTCATTAGTTCTGGCACCAGC	60	108	AW289395
	R-CACACGAAGGTGATGAACATG			
IFNAR1	F-GCGAAGAGTTTCCGCAACAG	57	275	NM_174552.2
	R-TCCAAGGCAGGTCCAATGAC			
IFNAR2	F-TCGTATGTTGCGCCTGTTCT	56	231	NM_174553.2
	R-GTCCGTCGTGTTTACCCACA			
NFKβ2	F-CCTGCTGAATGCTCTGTCTG	58	102	NM_001102101.1
	R-TCCTCCTTCACCTCTGTGCT			
NFKβIA	F-AAGTGGTCCGCCAAGTGAAG	56	105	NM_001045868.1
	R-CGATTTCTGGCTGGTTAGTGATC			
TNFα	F-CCCCAGGGCTCCAGAAGTTGC	60	103	XM_005696606.3
	R-GGCCGATTACCCCGAAGTGC			
TGFβ1	F-CGCTCCTGTGGCTACTAATGCTGA	58	149	NM_001314142.1
	R-CGGGGGACTGGCGAGCC			
IL1B	F-AATCGAAGAAAGGCCCGTCT	58	51	DQ837159.1
	R-ATATCCTGGCCACCTCGAAA			
IL10	F-AGCCAGCCTGCCCCACAT	58	140	DQ837159.1
	R-TCCCCCAGCGAGTTCACG			
IL17	F-CACAGCATGTGAGGGTCAAAC	56	83	NM_001008412
	R-GGTGGAGCGCTTGTGATAAT			
PTGS2	F-CATGGGTGTGAAAGGGACGAAAGA	60	182	XM_018060731.1
	R-CCTTAGTGAAAGCTGGTCCTCGTT			
Cytokeratin	F-CCCCCAGGTCCTTCAGCAGCC	64	147	NM_001290975.1
	R-GGGCCCCACCGTAGCTTCCAG			
Vimentin	F-CCGACGCCATCAACACCGAGT	60	163	NM_173969.3
	R-TTGCCCTGGCCCTTGAGCTG			
GAPDH	F-GCGATACTCACTCTTCTACTTTCGA	58	82	U85042.1
	R-TCGTACCAGGAAATGAGCTTGAC			

## Data Availability

The data presented in this study are available on request from the corresponding author.

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
