# Peer review of "Embryo–Uterine Cross-Talk: Exploration of the Immunomodulatory Mechanism in Buffalo"

_animals, 2022, doi:10.3390/ani12223138_

Round 1

Reviewer 1 Report

The paper Embryo-Uterine Cross-Talk: Exploration of Immunomodulatory Mechanism in Buffalo by Lakshmi Devi and colleagues aims to understand the molecular cross talk between embryo and uterine endometrium. In particular, this work was undertaken to investigate the effect of pre-implantation embryo on the expression profile of immune-related genes in uterine epithelial cells and PBMCs in buffalo, thus this manuscript falls within the scope of ANIMALS and the special issue Early Embryo Development in Agricultural Animals.
The manuscript is correctly prepared with clear material & methods and results. well-summarized conclusions. The introduction provide sufficient background with related bibliography and the conclusions are suppoted by the results.
There are only few minor comments to adress.
The sentences to be checked are highlighted in yellow inside the manuscript file.

Introduction
Lines 79-82: Please recheck the sentence, in particular the tense of “confers”. I don't feel qualified to judge about the English language, buti t sound strange to me, may be it would be “confering”.
Material and Methods
Lines 236-241: Please check the reference to the paragraph in which is explained the method for the in vitro embryo production. It would be 2.1 instead of 2.3. Moreover, it would be better to explain what do you mean for 1st IVC, 2nd IVC etc. I suppose that are the time in which the colture media are changed. At this regard, how often were they changed?
Lines 270-278: The description of Gene expression analysis (2.3.6) is the same reported in the paragraph 2.2.3. I think it could be referred to that description.

Author Response

Thanks for kind review and critical comments. Point wise response is as follows:

Comment 1

Introduction
Lines 79-82: Please recheck the sentence, in particular the tense of “confers”. I don't feel qualified to judge about the English language, but it sound strange to me, may be it would be “confering”.
Response: Word Confer has been used for “the embryo” so it is justified and has not been changed. Pl.

Material and Methods

Comment 2

Lines 236-241: Please check the reference to the paragraph in which is explained the method for the in vitro embryo production. It would be 2.1 instead of 2.3. Moreover, it would be better to explain what do you mean for 1st IVC, 2nd IVC etc. I suppose that are the time in which the culture media are changed. At this regard, how often were they changed?

Response: Sorry, it was by mistake and has been corrected.  IVC is explained for better understanding.
Comment 3

Lines 270-278: The description of Gene expression analysis (2.3.6) is the same reported in the paragraph 2.2.3. I think it could be referred to that description.

Response: agreed and changed as suggested.

Reviewer 2 Report

This manuscript provides some interesting and important data relating to the immunological crosstalk between embryos and uterine epithelium. The studies that have been performed to this point appear well controlled and performed, with some clear relevance to the interaction between buffalo conceptus' and the uterus. While the work clearly demonstrates differences in the expression of mRNA transcripts encoding important immune regulating genes in uterine epithelium, this is not sufficient to support the authors conclusions on its own.

Major change:

1) The authors should perform some immunologic-based assays (ELISA, Western blot, immunofluorescence) on media (for soluble factors) and cells (for membrane associated factors) to determine whether the changes observed in mRNAs are reflected in the expression of functional proteins. This is important as there are many instances where changes at the transcript level are not reflected in protein. Studies should be done both for soluble factors and membrane/intracellular factors under the experimental conditions described. It may not be necessary to prove these changes for every factor identified at the mRNA level, however, validation of key changes at the protein level should be performed for several key factors identified, after which the authors could credibly claim that other molecules that participate in embryo/uterine crosstalk behave similarly. This is indispensable for any assumptions regarding the biological roles of these factors in the crosstalk mechanisms.

Author Response

Thanks for kind review and critical comments. Pointwise response is as follows:

Comment 1

The authors should perform some immunologic-based assays (ELISA, Western blot, immunofluorescence) on media (for soluble factors) and cells (for membrane associated factors) to determine whether the changes observed in mRNAs are reflected in the expression of functional proteins. This is important as there are many instances where changes at the transcript level are not reflected in protein. Studies should be done both for soluble factors and membrane/intracellular factors under the experimental conditions described. It may not be necessary to prove these changes for every factor identified at the mRNA level, however, validation of key changes at the protein level should be performed for several key factors identified, after which the authors could credibly claim that other molecules that participate in embryo/uterine crosstalk behave similarly. This is indispensable for any assumptions regarding the biological roles of these factors in the crosstalk mechanisms.

Response: We do also agree that studying expression of genes by protein expression is gold standard but due to some constraints it could not been done. However, several other studies proved that these molecules come in secretion. We will certainly take care in our future studies.

Reviewer 3 Report

This interesting study helps on the elucidation of maternal recognition mechanism in buffalo by studying some imunomodulatory patterns of embryo and uterine cells response. It can be accepted after only minor revisions.

1. Tittle - Ok

2. Summary - It is well written, but authors should link it with their own experiment, highlighting their main findings in a coloquial language.

3. Abstract - Well written.

4. Keywords - Try to use indexing terms not previously used mainly in the tittle and abstract. For example, include maternal recognition, embryo development, etc.

5. Introduction - Even if this section is well-written and presents a large basis for the conduction of the experiment, authors are advised to clearly report the specific aims of the study in the final paragraph.

6. Methods - Please inform how many samples (from how many individuals) were used in the experiment.

- Please provide references for all the methods, for example for immunohistochemistry.

- In experiment 1, how many embryos were used per group?

- Regarding statistical analysis, wouldn't a test for Frequency distribution like Chi-square  be more appropriate than ANOVA for this kind of data?

7. Results - The first sentence is unnecessary; there is no need to remember the aims of the study at this point. Results are clearly presented and figures are really good.

8. Discussion - At general, authors well-discussed the main findings of the study and provide objective conclusions.

9. References - ok

Author Response

Thanks for kind review and critical comments. Pointwise response is as follows:

Comment 4

Keywords - Try to use indexing terms not previously used mainly in the title and abstract. For example, include maternal recognition, embryo development, etc.

Response: agreed and changed as suggested.

Comment

Introduction - Even if this section is well-written and presents a large basis for the conduction of the experiment, authors are advised to clearly report the specific aims of the study in the final paragraph.

Response: agreed and tried to make it more understandable.

Comment 6.

Methods - Please inform how many samples (from how many individuals) were used in the experiment.

- Please provide references for all the methods, for example for immunohistochemistry.

- In experiment 1, how many embryos were used per group?

- Regarding statistical analysis, wouldn't a test for Frequency distribution like Chi-square be more appropriate than ANOVA for this kind of data?

Response: Sample size has been clarified in the text and references have been inserted as suggested. For statistical analysis, as the data has been arsine corrected, this can be analyzed using ANOVA as other reports are also there including ours.

Comment 7. Results - The first sentence is unnecessary; there is no need to remember the aims of the study at this point. Results are clearly presented and figures are really good.

Response: Changed as suggested.

Comment 8. Discussion - At general, authors well-discussed the main findings of the study and provide objective conclusions.

Response: Changed as suggested.

Round 2

Reviewer 2 Report

My previous suggestion was to provide some protein-based analysis (IHC, Western Blot) to verify that the molecular changes in mRNA levels were reflected in some functional proteins (I didn't even request that they validate all genes). It is unfortunate that the authors are unable to do so, especially considering their use of IHC for other aspects of the study (which suggests they have the ability and resources to employ these techniques.   At the very least, the authors should describe this limitation in the discussion of their paper, and cite/highlight examples from other work in the embryo/uterine environment where comparable changes occur (a simple PubMed search provides many recent examples in bovine).   Although the English has been slightly improved, some further refinement would be helpful.